# Coronary Vasculitis

**DOI:** 10.3390/biomedicines9060622

**Published:** 2021-05-31

**Authors:** Tommaso Gori

**Affiliations:** Kardiologie I and DZHK Standort Rhein-Main, Universitätsmedizin Mainz, 55131 Mainz, Germany; Tommaso.gori@unimedizin-mainz.de

**Keywords:** coronary artery disease, vasculitis, inflammatory diseases

## Abstract

The term coronary “artery vasculitis” is used for a diverse group of diseases with a wide spectrum of manifestations and severity. Clinical manifestations may include pericarditis or myocarditis due to involvement of the coronary microvasculature, stenosis, aneurysm, or spontaneous dissection of large coronaries, or vascular thrombosis. As compared to common atherosclerosis, patients with coronary artery vasculitis are younger and often have a more rapid disease progression. Several clinical entities have been associated with coronary artery vasculitis, including Kawasaki’s disease, Takayasu’s arteritis, polyarteritis nodosa, ANCA-associated vasculitis, giant-cell arteritis, and more recently a Kawasaki-like syndrome associated with SARS-COV-2 infection. This review will provide a short description of these conditions, their diagnosis and therapy for use by the practicing cardiologist.

## 1. Introduction

The term vasculitis refers to a group of conditions whose pathophysiology is mediated by inflammation of blood vessels. Most forms of vasculitis are systemic and may present variable clinical manifestations, requiring a multidisciplinary approach. A number of etiologies have been reported; independently of the specific organ involvement, vasculitis can be primary or secondary to another autoimmune disease or can be associated with other precipitants such as drugs, infections or malignancy [1]. Almost all cases of coronary vasculitis appear as a manifestation of systemic (primary) vasculitis, which are classified based on the type and size of the vessels affected and the cellular component responsible for the tissue infiltration. The classifications of the American College of Rheumatology, and the Chapel Hill International Consensus Conference distinguish among the following groups of vasculitis [2]: (a) large vessels vasculitis with giant cell (temporal) arteritis and Takayasu arteritis; (b) medium-sized vessels vasculitis including polyarteritis nodosa and Kawasaki disease; (c) small vessel vasculitis including allergic granulomatous angiitis, Wegener’s granulomatosis, microscopic polyangiitis (microscopic polyarteritis), Henoch–Schonlein purpura, essential cryoglobulinemic vasculitis, and cutaneous leukocytoclastic angiitis [2]. Cardiac manifestations of systemic vasculitis are relatively rare but may lead to a sudden and rapid worsening of the patient’s prognosis [3,4], including sudden death in young individuals with no prior cardiovascular disease. Beyond coronary artery vasculitis, clinical manifestations of cardiac involvement may include pericarditis, myocarditis, heart valve disease. Coronary vasculitis may present as stenosis, occlusion, aneurysm or rupture of coronaries. Although extremely rare, coronary vasculitis should be considered in the differential diagnostic of young patients with otherwise unexplained acute coronary syndromes or congestive cardiac failure and in patients with known primary or secondary vasculitis. Among the different forms listed above, the most frequent forms of coronary vasculitis include polyarteritis nodosa, Kawasaki’s disease, Takayasu’s arteritis and giant cell arteritis; most recently, an inflammatory involvement in patients infected with the SARS-COV-2 virus has also been reported. Early diagnosis of these rare forms has a drastic impact on the diagnostic methods and the choice of therapy [5], and appropriate treatment has the potential to modify the severity of the disease.

## 2. Most Common Vasculitis Entities with Coronary Involvement

Although a thorough discussion on the clinical features, pathophysiology and therapy of vasculitis goes beyond the scope of this paper focusing on coronary involvement, the following paragraphs will describe shortly the main features of each condition associated with coronary involvement, also summarized in Table 1.

### 2.1. Large Vessel Vasculitis

#### 2.1.1. Takayasu

Takayasu arteritis affects more commonly young females and it involves large vessels, typically the aortic arch and proximal branches of the aorta as well as the pulmonary arteries. The course of disease is usually chronic, with cutaneous, neurologic, gastrointestinal, and constitutional symptoms. Angina pectoris may occur following coronary artery ostial stenosis from aortitis or coronary arteritis in 10–45% of the cases and may have severe clinical sequelae, even though regression upon immunosuppression is possible [19]. In these cases, coronary artery stenosis is caused by the diffusion of the inflammatory process and intimal proliferation form the wall of the ascending aorta [6]. The pathogenesis is unclear, but cell-mediated mechanisms (particularly cytotoxic lymphocytes, especially gamma delta T lymphocytes releasing the cytolytic protein perforin) are thought to be involved [20]. The presence of a (yet unknown) specific antigen in the aorta might explain the selective localization of the disease [21]. At histology, mononuclear cells—predominantly lymphocytes, histiocytes, macrophages, and plasma cells—are present, and giant cells and granulomatous inflammation are found in the arterial media. The expansion of these inflammatory processes may lead on one side to destruction of the elastic lamina, causing the aneurysmal dilation of the affected vessel, or to narrowing of the lumen. Systemic inflammatory markers are often elevated. Systemic therapy consists of corticosteroids and immunosuppression, but anti-IL6 therapy with Tocilizumab has also been associated with a reduction in the markers of inflammation and, importantly, also the number of coronary artery lesions [22]. Takayasu is a progressive chronic disease with remissions and relapses and an 80–90% survival at five years [23].

#### 2.1.2. Giant Cell Arteritis 

This condition shares several similarities with Takaysu’s arteritis, and the differential diagnostic is based on a rather artificial criterium of age (older than 50 in giant cell arteritis and younger than 40 in Takayasu) and vascular district involvement (more frequent involvement of supraaortic vessels in giant cell arteritis) [24]. Both giant cell arteritis and Takayasu involve the aorta and its major branches and are indistinguishable histopathologically. Although the prevalence of coronary artery involvement in giant cell arteritis is thought to be lower [8], large cohort studies report that the incidence of acute myocardial infarctions is twice as high as that in patients without giant cell arteritis, particularly early after diagnosis (Hazard Ratio for AMI: 11.9 [2.4–59.0]) [25]. Therapy with tocilizumab, an anti-IL6 antibody, is recommended in addition to corticosteroids, but its efficacy specifically on coronary vasculitis has not been investigated. The disease does not compromise life expectancy except for those patients in whom aortic dissection is present [8]. 

### 2.2. Medium Vessel Vasculitis

#### 2.2.1. Polyarteritis Nodosa

Polyarteritis nodosa is a systemic necrotizing vasculitis which predominantly involves medium-sized arteries, more rarely small muscular arteries in middle-aged adults. Unlike microscopic polyarteritis, granulomatosis with polyangiitis, polyarteritis nodosa is antineutrophil cytoplasmic antibodies (ANCA)-negative [2]. It may affect virtually any organ in the body (kidneys, skin, joints, muscles, nerves, and gastrointestinal tract), usually in combination, but usually does not involve the lungs. Coronary involvement is considered to be rare but may be severe, including aggressive three-vessel disease with infaust outcome (Figure 1) [26]. Clinical sequelae of this involvement include congestive heart failure, hypertension, pericarditis, and arrhythmias. 

Coronary involvement may manifest in the form of stenosis, occlusion, aneurysm, or dissection. In a recent review of cases, a total of 34 patients with an average age of 41 years were identified from 32 publications. Male sex is more frequent, coronary disease was the first manifestation of polyarteritis nodosa in ¾. The clinical course of the disease was in general very severe, with cases of death from cardiac arrest, pulmonary edema with alveolar hemorrhage or multiple intracranial hemorrhages after thrombolytic therapy [27]. The formation of immune complexes as the result of a virus infection (hepatitis B or C virus) and hairy cell leukemia is thought to mediate the inflammatory reaction, which most often lead to media thickening and stenosis rather than aneurysm formation. Medical therapy includes cyclophosphamide in addition to corticosteroids and/or azathiprine, even though the long-term effectiveness of this approach appears to be limited. Five-year survival reaches 80%. 

#### 2.2.2. Kawasaki

Also known as mucocutaneous lymph node syndrome, Kawasaki disease is an acute, self-limited small and medium vessel systemic vasculitis with a frequent involvement of coronary arteries (right more often than left) affecting children <5 years of age with rare cases in older children [28]. Although Kawasaki disease generally has a self-limiting febrile course, given the decreasing incidence of rheumatic disease, it represents the most common cause of acquired heart disease in childhood in developed countries [29] and it accounts for as many as 5% of the cases of acute coronary syndrome in patients younger than 40 years. Clinically, it presents as polymorphous rash, mucosal changes (including dry, cracked lips and strawberry tongue), extremity changes (including palmar and/or plantar erythema, swelling, and desquamation), bilateral nonpurulent conjunctivitis, and cervical lymphadenopathy (≥1.5 cm diameter), usually unilateral. Although a genetic influence has also been hypothesized, like for Behçet, the pathogenesis of Kawasaki is not clear and may be related to a wind-borne or water-borne pathogen. The most commonly affected age group are children under five years of age, but cases in adults are also common. For instance, a case of adult-onset Kawasaki Disease Shock Syndrome complicated by coronary aneurysms in a 20-year old man of East Asian ancestry has been recently reported [30]. Like for Behçet, the prevalence is higher, and the prognosis worse, in males affected by Kawasaki [12]. In the current hypothesis, it is believed that the activation of an immune reaction involving in the first phase neutrophils and then lymphocytes, cytokines, and proteinases; tumor necrosis factor alpha (TNF-a); Interleukin 1, 4, and 6 and matrix metalloproteinases (MMP3 and MMP9) triggered by exposure to an airborne virus may subtend Kawasaki. CD8+T cells, plasma cells, and monocytes cause release of pro-inflammatory cytokines IL-1β and TNFα. These processes may evolve for months to years resulting in a chronic arteritis. Oligoclonal IgA plasma cells appear to be central in the cascade leading to coronary arteritis. Clinical manifestations may include myocarditis and arteritis resulting in fibrinoid necrosis of the internal elastic lamina and subsequent formation of coronary aneurysms in up to one-third of untreated patients. Cerebral aneurysms are less frequent at 1–2% of patients. Monocytes, neutrophils and macrophages appear to be involved in the pathogenesis of these vascular lesions. Resulting from these inflammatory processes, an inappropriate healing response may also cause coronary stenosis. Otherwise, typical complications associated with the presence of the aneurysms include thrombus formation causing embolism and peripheral occlusion and rupture. Cases of regression of small coronary aneurysms upon inflammatory therapy have also been reported [31]. In the acute phase of Kawasaki disease, therapy with intravenous Immunoglobulin, corticosteroids, and aspirin monotherapy is encouraged [32]. Therapy with immunoglobulins and corticosteroids is associated with a reduction in the incidence of coronary events (odds ratio: 0.3 [0.2–0.5]) [33], but the presence of signs compatible with coronary vasculitis is a negative predictor for the responsiveness to immunoglobulins. The prognosis is variable and depends on the size of the aneurysms. Regular follow-up is therefore important. 

### 2.3. Small Vessel Vasculitis

#### Eosinophilic Angiitis

Anti-neutrophil cytoplasmic antibody (ANCA)-associated vasculitides compose a family of conditions involving severe, systemic, small-vessel vasculitis featuring autoantibodies directed against the neutrophil proteins leukocyte proteinase 3 or myeloperoxidase. Their clinical presentation and therapies have been recently reviewed [34]. Within this group of diseases, eosinophilic granulomatosis with polyangiitis most commonly involves coronary vessels. This condition (which is not necessarily mediated by ANCA) has an incidence of ~0.5–2 cases per million per year, its time of onset is typically in the middle or older age and is equally frequent in males and females. Histologically, it is characterized by eosinophil-rich and necrotizing granulomatous inflammation predominantly affecting small-to-medium vessels; eosinophilia is the typical marker and pulmonary involvement and asthma are often associated. Connected, but not overlapping, with this condition, the first case of a novel medium-sized arteritis with eosinophilic inflammation limited to the adventitia and periadventitial soft tissue of epicardial coronary arteries was reported in 1989. Eosinophilic coronary periarteritis is not systemic and differs from poliarteritis nodosa and Churg–Strauss Syndrome in that it does not involve other arterial layers and it does not show fibrinoid necrosis. This form of coronary artery vasculitis has an adult onset and it has been associated with cases of spontaneous coronary artery dissection or coronary spasm [35,36]. Eosinophilic arteritis without dissection is more frequent in men than women, while a spontaneous dissection—prevalently of the left anterior descending coronary—is more frequent in females. Although the etiology and pathogenesis of this condition remain uncertain, it is possible that this disease may play a role in several cases of spontaneous dissection that remain etiologically undiagnosed. Churg–Strauss syndrome is a related disease that is diagnosed in the presence of the following six criteria: severe asthma, fleeting pulmonary infiltrates on chest radiographs, eosinophilia, paranasal sinus abnormalities, eosinophilic infiltration on biopsy, and neurologic manifestations. The highest incidence is in the fourth-fifth decade of life and is higher in females. Cardiac involvement may manifest as pericarditis, restrictive or dilated cardiomyopathy, myocarditis, arrhythmias, and sudden cardiac death. The prognosis of these diseases is markedly improved by the use of corticosteroids, and the mortality is in the range of 30% at 5 years [37].

### 2.4. Other Forms

#### 2.4.1. Behçet

Behçet disease is a severe disease most frequent in the eastern Mediterranean and the east, where it reaches an incidence of 0.03% of the population. The incidence in males, most commonly in the fourth and fifth decade of life, is three times higher than in females. The most common manifestations include signs of inflammatory activation such as fever and constitutional symptoms. Clinical features of this disease include oral and/or genital ulcerations, ocular disease, and arthritis. Behçet´s disease has an unknown cause even though a genetic predisposition is suspected; its pathophysiology is based on autoinflammatory mechanisms, endothelial damage/dysfunction, and impaired fibrinolysis involving the activation of T-lymphocytes, including T helper 17 cells, immune complex formation, neutrophil activation, and the secretion of inflammatory cytokines. From the histologic perspective, Behçet is associated with perivasculitis with neutrophil infiltration, fibrinoid necrosis and endothelial swelling [15,16]. Anti-lymphocyte and anti-cardiolipin antibodies are a feature of this condition and allow its diagnosis. Arterial involvement in Behçet disease manifest itself as aneurysms of the medium- and large-sized arteries. The vascular manifestations of Behçet vary between 10 and 30% of the cases and may involve both arteries and, approximately 4 times more frequently, veins [14]. Vascular involvement portends a negative prognosis, particularly in younger males without traditional coronary artery disease risk factors (except for smoking). Although native coronary disease is most commonly described, cases of in-stent restenosis have also been reported [14]. Therapy may include colchicine and general immunosuppressive therapy with corticosteroids, azathioprine, cyclosporine A, and cyclophosphamide). The prognosis is good, although the incidence of relapses is very high. 

#### 2.4.2. Erdheim–Chester Disease

Erdheim–Chester disease is a rare non-Langerhans histiocytosis and is essentially a malignancy of myeloid progenitor cells caused by a somatic mutation of signaling molecule genes, which explains the increased expression of inflammatory cytokines (IL-6, interferon alpha, MCP-1). Fewer than 1000 cases of this disease have been reported, most commonly under the form of multifocal sclerotic lesions of the bones. Bone pain, neurologic symptoms, and cardiac symptoms are the most common manifestations. It may result in multisystem involvement including vasculitis-like phenomena of large and medium-sized arteries, particularly of the aorta (around 60% in a recent case series) [17]. Coronary artery involvement is rather common (55%), leading to about one-third of deaths [38]. The most common presentation includes infiltration, particularly at the level of the right atrium; periarterial fibrosis, thickening and pleural effusion are also common. Its histologic hallmark is perinephric fibrosis. The treatment may include a BRAF inhibitor (vemurafenib), a MEK inhibitor, interferon alfa, or immunosuprression with corticosteroids or cytotoxic therapies. Five-year survival reaches 70%.

#### 2.4.3. IgG4-Related Disease

Immunoglobulin G4-related disease is a rare fibroinflammatory condition with multiorgan lymphoplasmacytic infiltration that affects more comonly males. Every organ may be affected and extracardiac involvement may manifest as sialadenitis, thyroiditis, nephritis, lymphadenopathy, and lung disease. The pathogenesis of this disease appears to depend on an immune response mediated by Th1 and Th2 cells, while the role of IgG4 antibodies remains unclear. Cardiovascular manifestations of IgG4-RD may include aortitis, medium-sized vessel arteritis, pulmonary vascular disease, phlebitis, valvulopathy, pericarditis, and antineutrophilic cytoplasmic antibody-associated vasculitis and more frequently involve the abdominal vasculature, even though cardiac cases have also been reported [39,40]. Therapy with glucocorticoids leads to a remission in 98% of the cases, but a spontaneous remission without therapy is also often observed. Cases that are refractory to steroids and require immunosuppression, however, also exist. 

## 3. Epidemiology

The relative rarity of these conditions as well as a lack of standardized prospective imaging studies systematically evaluating the coronary vasculature complicates the diagnosis of coronary artery vasculitis in patients with known autoimmune diseases. The involvement of the coronary microvasculature (not visible at computed tomography or angiography) and the fact that coronary artery vasculitis may manifest under the form of coronary (micro)vascular spasm rather than fixed stenosis further complicate the diagnosis and may result in many false negatives. Finally, the co-existence of accelerated atherosclerosis caused by immunosuppressive therapies and the lack of availability of methods for the differential diagnosis of these two forms of disease is an additional hurdle to the diagnosis. Particularly in cases of isolated coronary artery vasculitis, which is considered to be extremely rare, the true incidence is probably dramatically underestimated [10]. A coronary artery vasculitis is diagnosed (Table 1) in 50% of the patients with polyarteritis nodosa (with a reported incidence of 4–10 per million per year) and in about 20% of patients with Kawasaki disease (incidence 2 per million per year [5]). A similar incidence of coronary artery vasculitis is reported for patients with large vessel vasculitides, such as Takayatsu arteritis and giant cell arteritis (incidence rates of 1–3 per million each). Other forms of large vessel vasculitides like the Erdheim–Chester disease are rarely associated with coronary artery vasculitis (about 5% of the cases). The incidence of vascular involvement in Behçet has also been reported to be in the range of 50% [41].

## 4. Pathogenesis and Pathology

The pathogenesis of coronary artery vasculitis is complex, multifaceted, depends on the specific disease, and it is essentially mediated by both extrinsic and host factors such as immuno-mediated inflammation and auto-antibody dependent reactions [9]. Inflammatory cytokines, in part specific for each condition, including interferon-gamma (IFN-γ), tumor necrosis factor-alpha (TNF-α), and T-1 interleukins have all been reported. Histopathological findings on autopsy are also specific for each condition. The coronary artery vasculitis caused by polyarteritis nodosa is characterized by intramural, perivascular lymphocyte, and macrophage infiltration as well as fibrinoid necrosis which provoke the destruction of the arterial wall. Takayatsu and giant cell arteritis are associated with intimal hyperplasia, granulomatous arteritis, and coronary atherosclerosis. In contrast, Kawasaki disease is characterized by multi-cellular infiltration of the arterial wall causing necrosis of the internal elastic lamina. As a result of these different mechanisms, findings of chronic inflammation, scar tissue, necrosis, and stenosis may be present in all forms of coronary artery vasculitis. Coronary artery aneurysm are a rarer entity caused, as mentioned above, by weakening of the arterial wall mediated by an overactivity of metalloproteinases and metalloelastases. The same mechanisms, coupled with the release of pro-angiogenetic substances from the eosinophils and the resulting formation of weak capillaries that may rupture in the arterial media, may lead to spontaneous dissection in eosinophilic arteritis [42]. Arterial thromboses may result from the blood stasis in these aneurysms and/or in stenosis leading to myocardial ischemia [43]. As in all inflammatory conditions, the incidence of typical coronary atherosclerosis is also higher in patients with vasculitis. Rarely, embolisms of Wegener granulomatosis of heart valve lesions causing coronary artery occlusion have also been described [44].

## 5. Clinical Manifestations and Diagnosis

The symptoms and signs associated with system involvement differ significantly among different disease entities reflecting the different organs involved. Because of the relative rarity of these disease, a diagnosis can easily be missed. The clinical presentation may not be specific, but clinical features that might suggest an inflammatory etiology include elevated inflammatory markers that cannot be explained otherwise, signs of inflammation—such as fever, chills, night sweats, weight loss, subclavian or aortic bruits—suggestive of atypical stenosis in other districts, a history of multiple system involvement such as abdominal ischemic events, early age particularly in the absence of genetic predisposition or severe risk factors. Patients with suspected vasculitis, particularly Takayasu, should undergo imaging of the arterial tree by magnetic resonance angiography (which has the advantage of avoiding contrast medium and radiation exposure) or computed tomography to evaluate the presence of aneurysms, stenoses, arterial wall thickening, and/or masslike lesions surrounding the coronary arteries as recently reviewed in [45,46]. In particular, computed tomography provides accurate non-invasive information on both luminal and mural lesions in the aorta and its branches, which is important in both the diagnosis and the follow-up of disease progression. Computed tomography allows detection and quantification of stenoses and aneurysms which may be present in virtually every form of vasculitis [46]; findings that are typical of specific conditions include “skip lesions“ (focal stenoses in Takayasu disease) and “pigs-in-a-blanket” lesions (rings of soft-tissue attenuation surrounding the coronary arteries) in coronary periarteritis [18]. Ultrasound imaging of other districts, including the chest, abdomen, head, and neck may be useful, particularly to detect wall thickenings which cannot be diagnosed by angiography. Positron emission tomography, often in combination with computed tomography or magnetic resonance, has been used for the diagnosis of large-vessel vasculitis. Arterial segments featuring increased standardized uptake values may be suggestive of disease [47]; these abnormalities are however not present in coronaries.

At angiography, coronary artery vasculitis may manifest under the form of coronary stenosis, aneurysm (Figure 2), dissection (Figure 3, most frequent in eosinophilic periarteritis), spasm (Figure 4), or coronary rupture (Table 1). Sudden death, typical angina, acute myocardial infarction, atrial and ventricular arrhythmias, conduction disturbances, or cardiac failure have all been described [48]. Although no specific finding at coronary angiography allows a safe diagnosis, features associated with early, advanced, or atypical coronary artery disease may suggest an inflammatory etiology. For example, Kawasaki disease and polyarteritis nodosa are often associated with large aneurysms [11]. Polyarteritis nodosa often features multifocal aneurysms with a “beads on a string” or nodular appearance [49], which are also present in patients with anti-neutrophil cytoplasmic antibody (ANCA)-associated vasculitis and Behcet’s disease [13]. The arterial lesions in Behçet disease may be occlusive or aneurysmal [15]. Giant cell arteritis has been associated with long coronary lesions [50], and in Takayasu’s arteritis, coronary lesions have been classified in three main types: 1, stenosis or occlusion of the coronary ostia (60–80%); 2, diffuse disease that may involve all epicardial branches or only focal segments (10–20%); and 3, coronary aneurysms (0–5%) [23].

Coronary angiography does not detect inflammation, necrosis or reactive thickening of the arterial wall. Non-invasive assessment of the cardiac and extracardiac circulation using computed tomography or magnetic resonance imaging is therefore recommended in patients with large (giant cell arteritis and Takayasu’s arteritis), medium (poliarteritis nodosa) or variable (Behçet’s disease) vessel vasculitis. Patients with small vessel vasculitis (i.e., ANCA-associated) often have associated myo- or pericarditis, which should also be investigated using magnetic resonance imaging. Periarterial soft tissue thickening or extrinsic compression are also features of more rare conditions such as Immunoglobulin G4(IgG4)-related disease or the Erdheim–Chester disease [18,51].

## 6. COVID-19 and Coronary Inflammation 

The impact of COVID-19 on vascular biology is a topic of recent great attention as it appears to be responsible for a significant percentage of the negative outcomes [52]. Macro- and microvascular thrombosis involving arteries, veins, arterioles, capillaries, and venules in all major organs has been reported, along with evidence of endotheliitis across different vascular beds [53,54]. The lesions featured diffuse lymphocytic endotheliitis and apoptotic bodies likely following virus adhesion to the cell and proinfammatory and apoptotic pathway signaling [55]. Oxidative stress, increased production and release of chemokines, cytokines, and byproducts of damage-associated molecular patterns appear to be responsible for the endotheliitis [56]. Neutrophil recruitment by products of vascular injury is mediated by several pathways including the PI3K/AKt/eNOS/NF-Kβ and ERK1/2/ P38 MAPK-activated protein kinases and leads to further inflammation via release of TNF-α, IL-1, and IL-8, and neutrophil-extracellular traps NETs [57]. The mechanisms leading to necrosis and apoptosis have been recently summarized in dedicated reviews [58]. COVID-19 associated cytotoxicity primarily find place within the microcirculatory system, leading to ultrastructural changes and vascular dysfunction. Furthermore, increased bioavailability of the vasoconstrictor Angiotensin II due to COVID-19-mediated depletion of its receptor ACE-2, is an acknowledged mechanism of endothelial cell dysfunction. The release of von Willebrand factors which follows endothelial damage and activation can in turn recruit and activate circulating platelets contributing to the enhanced production of pro-coagulants, inflammatory cytokines, and NETs. Type I and II myocardial ischemia may result. Along with thrombotic events, vasculitis thus occurs in COVID-19 but its role in the several reported cases of acute coronary syndrome on COVID-19 patients remains unclear [57,59,60]. 

### COVID and Kawasaki-Like Disease

As described above, Kawasaki disease is believed to result from an excess innate immune response to viral pathogens. Although the mechanism remains to be elucidated, the involvement of the stimulator of interferon genes (STING, a cytosolic DNA sensor and adaptor protein in type I IFN and nuclear factor(NF)-κB pathway, leading to hyper-coagulopathy via macrophage production of tissue factor) pathway has been proposed [61]. A similar situation may occur in COVID-19 infections, where SARS–COV-2 binding to ACE2 increases STING pathway activation. The resulting immune hyper-responses, decreased lymphocyte counts, increased monocyte populations that secrete cytotoxic cytokines and heightened B and T cell responses configure a Kawasaki-like disease that may result in toxic shock syndrome or multi-systemic inflammatory disease [62]. In as many as 25% of the children affected by COVID-19, coronary dilations have been reported, and several cases of COVD-19 myocarditis have been published [63]. In a recent survey of 149 patients, children with “Kawacovid syndrome” were significantly older and presented more frequently gastrointestinal and respiratory involvement. Cardiac involvement in the form of myocarditis was more common (60%) as in traditional Kawasaki, but coronary artery abnormalities were rare. About 40% of patients with Kawacovid presented hypotension/non-cardiogenic shock [52]. At cardiac magnetic resonance imaging, evidence of diffuse myocardial edema has been reported [64]. The clinical presentation of the resulting multisystem inflammatory syndrome in children (MIS-C) reflects that of Kawasaki, but differences between the two exist with regards to age (from early childhood to late adolescence as compared to early childhood in Kawasaki); more frequent lymphopenia and thrombocytopenia, cardiac ventricular stress including myocarditis, and coagulopathy in MIS-C, more frequent gastrointestinal involvement, myocarditis leading to cardiogenic shock, and a different cytokine profile (IL-6 and IL-8 in MIS-C, IL-1 in Kawasaki).

## 7. Revascularization Procedures in Coronary Artery Vasculitis

### 7.1. Chronic Coronary Syndromes 

All patients with known vasculitis should be treated with acetylsalicilic acid. With regards to the indication to revascularization, traditional methods for the assessment of vitality (low-dose dobutamine stress-echocardiography, scintigraphy, magnetic resonance imaging) and ischemia (exercise-electrocardiography, stress echocardiography or scintigraphy, etc.) maintain their validity. There is little data, anecdotal in nature, regarding the outcomes of interventions in patients with coronary arteritis; current American Heart Association guidelines recommend percutaneous interventions in patients with either a single vessel involvement or focal multivessel disease [65]. Percutaneous coronary interventions might however not provide adequate revascularization in cases of diffuse disease. Given the severity of these diseases and their rarity, no prospective randomized trial to determine the results of different revascularization methods (interventional versus by-pass surgery) has been conducted. In patients with Takayasu’s arteritis, a higher rate of target lesion failure has been reported after percutaneous coronary intervention as compared to surgery (odds ratio 7.4 [2.4–23.1], *p* = 0.01) [66], such that surgery is recommended in these patients after induction of immunosuppression [19]. The involvement of the ascending aorta is a clear factor that complicates venous and free arterial by-pass graft surgery. As well, the internal mammary artery may also be a target of disease in patients with Takayasu’s arteritis [7], which complicates by-pass grafting [67]. Scar healing might be compromised or slower in patients with active disease. Finally, coronary artery aneurysms can be occluded by coiling or implantation of covered graft stents, even though the risk of thrombosis and restenosis remains high. Aneurysm resection/thrombectomy and by-pass surgery also remain options [68,69].

### 7.2. Acute Coronary Syndromes

Patients with coronary aneurysms have an elevated risk of myocardial infarction. Given the young age, interventional therapy in this setting is often not feasible. In case of severe thrombosis occurring within the aneurysm, therapy with heparin, IIbIIIa antagonists and fibrinolytics has been reported also in pediatric age [70,71]. However, the outcomes of these procedures are not known. Regardless of the type of revascularization, antiplatelet therapy should be applied, and life-long antiplatelet therapy is to be recommended in all patients with coronary aneurysms [65]. Data on anticoagulation in patients with aneurysms are also very scarce, and this type of approach is discouraged by some authors for fear of progression of the aneurysms [72] while others report a reduced incidence of acute myocardial infarctions in patients treated with a combination of aspirin and warfarin [73]. As well, despite anticoagulation, thrombosis in giant aneurysms can occur owing to blood stasis and decreased wall shear stress [74]. Finally, immunosuppressive therapy is a mainstay of therapy, and cases of “spontaneous” regression of coronary artery stenoses have been reported [75]. Cardiac transplantation may be considered in patients deemed not suitable for revascularization.

## 8. Conclusions

Coronary artery vasculitis is rare, but still represent one of the most frequent causes of coronary artery disease in young patients. Its anatomical manifestations may include coronary artery stenosis, aneurysms, thrombosis, and spontaneous dissection; and its consequences may be severe. Even though prognosis of coronary vasculitis is poor, early diagnosis and therapy improve survival rates. Both non-invasive and invasive methods provide essential information in the diagnosis. Large-scale studies are now necessary to further investigate the incidence, diagnostic yield, and therapy of this rare and heterogeneous group of diseases.

## Figures and Tables

**Figure 1 biomedicines-09-00622-f001:**
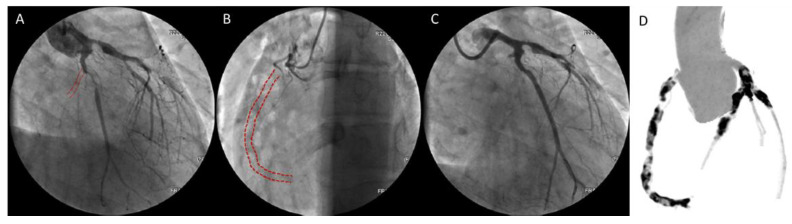
Coronary angiography (**A**–**C**) and computed tomography image (**D**) of three-vessel disease in a 22-years old patient with polyarteritis nodosa. Angiography showed chronic total occlusion of the right and circumflex coronaries. Reproduced with permission from [26] under the creative common license (http://creativecommons.org/publicdomain/zero/1.0/) (accessed on 3 March 2021).

**Figure 2 biomedicines-09-00622-f002:**
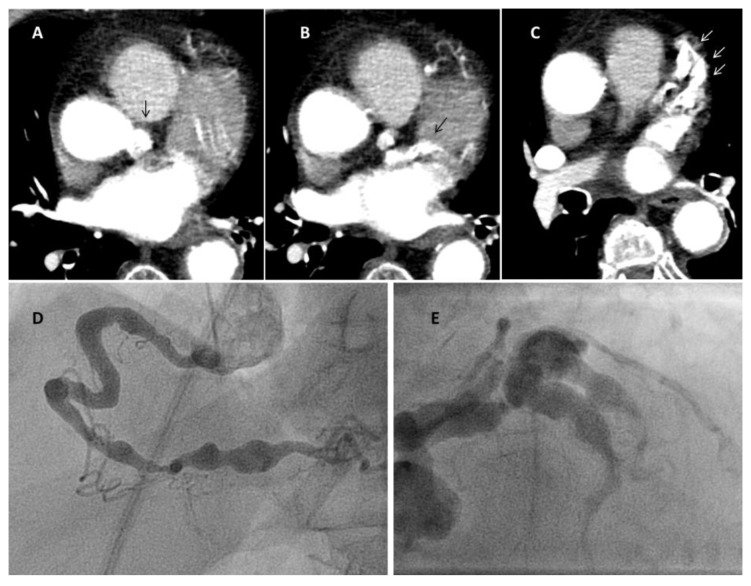
Coronary aneurysms in an 80-year-old female patient with stroke and rheumatoid arthritis. Panel a: right coronary; Panel b: left coronary (arrows mark the aneurysms). The patient had been admitted for sudden loss of consciousness. Chest computed tomography (panels **A**–**C**, Appendix A) was performed for the suspect of lung embolism. The exam showed severe coronary aneurysms. At angiography (panel **D**,**E**), the aneurysms were so large that they could not be imaged in one single run despite large contrast volume.

**Figure 3 biomedicines-09-00622-f003:**
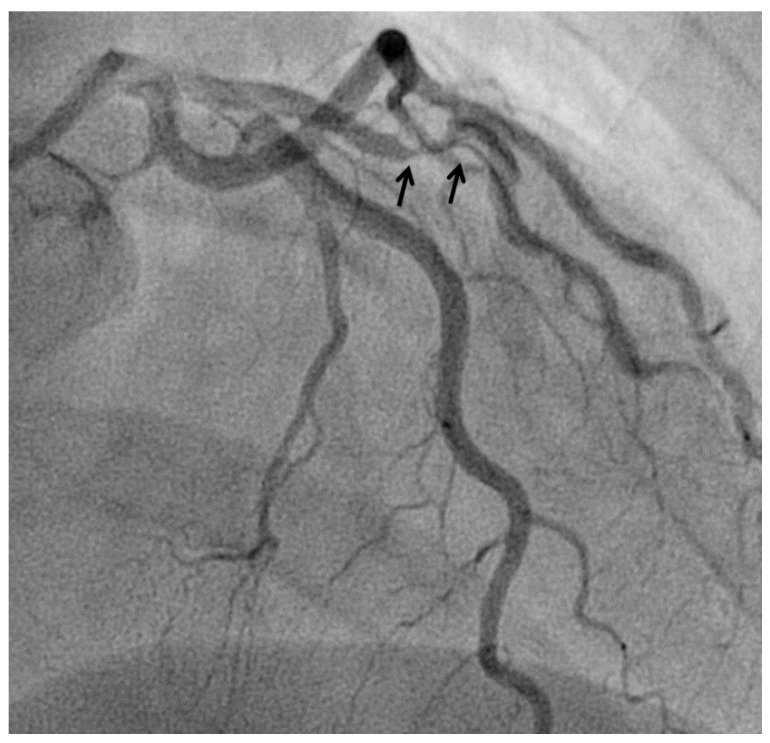
Spontaneous dissection (arrows) in a 45-years old woman causing non-ST elevation myocardial infarction.

**Figure 4 biomedicines-09-00622-f004:**
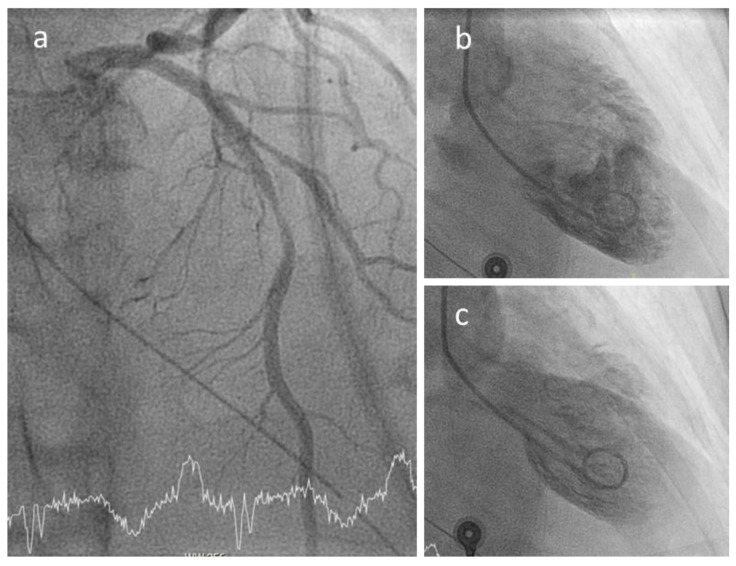
Coronary spasm in a 51-year-old female patient with ANCA-associated vasculitis and spasm of the proximal left anterior descending (panel **a**) causing a Tako-Tsubo-like contractile dysfunction (**b**,**c**).

**Table 1 biomedicines-09-00622-t001:** Typical forms of vasculitis with coronary involvement.

Group and Disease	Laboratory Findings	Frequency of Coronary Involvement	Location	Typical Lesion
Large Vessels				
Takayasu [6,7]	Inflammatory markers	10–45%	Ostial/proximal	Stenosis
Giant cell arteritis [8]	Inflammatory markers	Rare	Diffuse	Tapered smooth narrowings
Medium Vessels				
Polyarteritis nodosa [9,10]	Not consistently	10–50%	Not specific	Aneurysm or stenosis
Kawasaki [11,12]	Inflammatory markers	25–30%	Not specific	Aneurysm > stenosis
Small vessel				
Eosinophilic angiitis [10]	ANCA (not in all forms)	Rare	Not specific	Stenosis or aneurysm
Veins > arteries				
Behçet [13,14,15,16]	Not consistently	0.5–2%	Not specific	Thrombosis or pseudoaneurysm
Variable				
Erdheim–Chester [17]	Rare	25–55%	Right coronary prevalent	Periarteritis
IgG4 [18]	IgG4	1–3%	Not specific	Aneurysm or periarteritis

## Data Availability

Not applicable.

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
