# Peer review of "Coronary Vasculitis"

_biomedicines, 2021, doi:10.3390/biomedicines9060622_

Round 1

Reviewer 1 Report

In the present review, authors concentrated on the forms,  pathology and clinical manifestations of coronary vasculitis.

I have several comments and suggestions that might help improve the paper:

  1. The organization of the information for each form is different. For example, in "2.4 ANCA-associated vasculitis" information about therapy is missing but it is discussed in other forms. Moreover, in other forms (for example "Erdheim-Chester disease, IgG4-related disease" too little information is provided.
  2. There are many grammatical errors and the style is not very well organized.
  3. References are missing from Table 1 and figures 2,3,4
  4. Conclusions are missing
  5. Minor: please change the word "COVID-SARS" into "SARS-COV-2" in introduction.

Author Response

  • The organization of the information for each form is different. For example, in "2.4 ANCA-associated vasculitis" information about therapy is missing but it is discussed in other forms. Moreover, in other forms (for example "Erdheim-Chester disease, IgG4-related disease" too little information is provided.

You are absolutely right, I apologize. I have significantly modified paragraphs 2-1-2.7 to include all information also present in Table 1 (see also comment from reviewer 3) and to standardize the format. Of course some of these conditions are rare or very rare, and other ones are a spectrum of diseases rather than one signle entity, which complicates harmonizing the discussion. Also, my intent was to focus on the coronary involvement, but of course I agree that it is important to provide some general information about the disease.

  • There are many grammatical errors and the style is not very well organized.

I apologize. The paper has been entirely revised.

  • References are missing from Table 1 and figures 2,3,4

Thank you very much. I have added exemplary references to Table 1 (this table summarizes the findings in each type of vasculitis, so there is no single source). Figures 2, 3, 4 are original, there are no sources to reference.

  • Conclusions are missing

Thank you very much, a short paragraph 8 has been added.

  • Minor: please change the word "COVID-SARS" into "SARS-COV-2" in introduction.

This has been done, thank you

I hope you will find these changes satisfactory.

Reviewer 2 Report

The manuscript is interesting, it needs of a minor revision essentially for language, and for improving the style of Table 1 and the resolution of Figures (why are they marked in yellow ????). In addition, the review needs of a conclusion or general discussion 

Author Response

The manuscript is interesting, it needs of a minor revision essentially for language, and for improving the style of Table 1 and the resolution of Figures (why are they marked in yellow ????).

Thank you very much for your positive comments. I have reformatted Table 1 and added references (as suggested by reviewer 1). Higher resolution figures are available to the publisher for the final version of the manuscript. Legends are marked in yellow to facilitate identification for the editor. This yellow marking will disappear in the final published version should the paper be accepted. I also added a few figures presenting CT findings as suggested by reviewer 3.

In addition, the review needs of a conclusion or general discussion 

You are right. A short paragraph 8 has been added.

I hope these changes will find your agreement and look forward to your response.

Reviewer 3 Report

This is a review article on coronary vasculitis covering a range of disease from large vessels to small vessels and veins. Overall, the manuscript is nicely written. My main concern lies in the lack of imaging part, as imaging diagnosis, in particular coronary CT angiography is widely used as a less-invasive modality in the diagnosis of coronary artery disease. Authors should include some discussion of its value or value in this aspect. A nice review article entitled CT coronary angiography findings in non-atherosclerotic coronary artery diseases, Clin Radiol 2018;73:205-213 summarises coronary CT findings, especially there is a section on Coronary arteritis. I strongly suggest that authors consider adding something related to coronary CTA in the relevant sections. 

Specific comments:

  1. Page 2, Section 2, table 1 provides a summary of these diseases, but I think it is necessary to describe some details in the main text. This will lead to the following subsections such as 2.1, 2.2, etc.
  2. Page 5, Figure 1 seems strange as it does not look like the usual CT angiography image. Please double check it or was it presented in another format or windowing? Also computer tomography should be computed tomography. Same page, coronary involvement may be manifest in the form of.. grammatical error-may manifest in the form..
  3. Page 9, references should be cited to support the statement of non-invasive imaging assessment including PET imaging. See my suggest in the general comments about citing roles of coronary CTA. 
  4. Page 10, section 7.1, current American Heart Association... should be in the same paragraph, most likely due to formatting issue. 
  5. A brief summary and conclusion should be provided in the end after 7.2. 

Author Response

- This is a review article on coronary vasculitis covering a range of disease from large vessels to small vessels and veins. Overall, the manuscript is nicely written. My main concern lies in the lack of imaging part, as imaging diagnosis, in particular coronary CT angiography is widely used as a less-invasive modality in the diagnosis of coronary artery disease. Authors should include some discussion of its value or value in this aspect. A nice review article entitled CT coronary angiography findings in non-atherosclerotic coronary artery diseases, Clin Radiol 2018;73:205-213 summarises coronary CT findings, especially there is a section on Coronary arteritis. I strongly suggest that authors consider adding something related to coronary CTA in the relevant sections. 

You are absolutely right, thank you very much. Computed tomography is an asset of major importance for the diagnosis of coronary involvement.This is exemplified by the case presented in Figure 2 (CT scan pictures and more clinical details have been added): the patient was admitted for sudden loss of consciousness; a chest CT scan was performed for the suspect of lung embolism, the coronary aneurysms were actually a (clinically very relevant) incidental finding. I have expanded the discussion on CT and quoted the excellent papers by Broncano and Zhuo as references in this specific field.

- Specific comments:

Page 2, Section 2, table 1 provides a summary of these diseases, but I think it is necessary to describe some details in the main text. This will lead to the following subsections such as 2.1, 2.2, etc.

Thank you very much, you are right. Of course the quantity of information available (also given the differences in the incidence of each condition) is very different among different forms of vasculitis. I have modified significantly the paragraphs 2-1 to 2.7 to follow the same structure and include the information from Table 1.

- Page 5, Figure 1 seems strange as it does not look like the usual CT angiography image. Please double check it or was it presented in another format or windowing?

Thank you for this comment, I absolutely agree, this is a computer reconstruction. Unfortunately this case is not from our center (as reported in the legend), and this is the only CT image available. I have added the corresponding angiography.

- Also computer tomography should be computed tomography.

- Same page, coronary involvement may be manifest in the form of.. grammatical error-may manifest in the form..

Sorry, this mistake has been corrected.

- Page 9, references should be cited to support the statement of non-invasive imaging assessment including PET imaging. See my suggest in the general comments about citing roles of coronary CTA. 

Thank you very much. Teh following reference has been added: Grayson PC, Alehashemi S, Bagheri AA, Civelek AC, Cupps TR, Kaplan MJ, et al. (18) F-Fluorodeoxyglucose-Positron Emission Tomography As an Imaging Biomarker in a Prospective, Longitudinal Cohort of Patients With Large Vessel Vasculitis. Arthritis Rheumatol. 2018;70(3):439-49.

Page 10, section 7.1, current American Heart Association... should be in the same paragraph, most likely due to formatting issue. 

This has been corrected, thank you

- A brief summary and conclusion should be provided in the end after 7.2. 

I agree. A short paragraph 8 has been added.

Round 2

Reviewer 1 Report

The authors have clearly improved the manuscript. 

Reviewer 3 Report

Authors have done a great job in revising the manuscript. All of my comments are addressed in the revision. Thank you for your nice review article.